# A QIL1 Variant Associated with Ventricular Arrhythmias and Sudden Cardiac Death in the Juvenile Rhodesian Ridgeback Dog

**DOI:** 10.3390/genes10020168

**Published:** 2019-02-21

**Authors:** Kathryn M. Meurs, Steven G. Friedenberg, Natasha J. Olby, Julia Condit, Jess Weidman, Steve Rosenthal, G. Diane Shelton

**Affiliations:** 1Department of Clinical Sciences, College of Veterinary Medicine, North Carolina State University, Raleigh, NC 27607, USA; njolby@ncsu.edu (N.J.O.); jgcondit@ncsu.edu (J.C.); 2Department of Veterinary Clinical Sciences, College of Veterinary Medicine, University of Minnesota, Saint Paul, MN 55108, USA; fried255@umn.edu; 3CVCA at Chesapeake Veterinary Referral Center, 808 Bestgate Road, Annapolis, MD 21401, USA; jess.weidman@cvcavets.com (J.W.); Steven.Rosenthal@cvcavets.com (S.R.); 4Department of Pathology, University of California San Diego, La Jolla, CA 92093, USA; gshelton@ucsd.edu

**Keywords:** QIL1, Rhodesian ridgeback, arrhythmia, mitochondrial cristae

## Abstract

The *QIl1* gene produces a component of the Mitochondrial Contact Site and Cristae Organizing System that forms and stabilizes mitochondrial cristae junctions and is important in cellular energy production. We previously reported a family of Rhodesian Ridgebacks with cardiac arrhythmias and sudden cardiac death. Here, we performed whole genome sequencing on a trio from the family. Variant calling was performed using a standardized bioinformatics approach. Variants were filtered against variants from 247 dogs of 43 different breeds. High impact variants were validated against additional affected and unaffected dogs. A single missense G/A variant in the *QIL1* gene was associated with the cardiac arrhythmia (*p* < 0.0001). The variant was predicted to change the amino acid from conserved Glycine to Serine and to be deleterious. Ultrastructural analysis of the biceps femoris muscle from an affected dog revealed hyperplastic mitochondria, cristae rearrangement, electron dense inclusions and lipid bodies. We identified a variant in the *Q1l1* gene resulting in a mitochondrial cardiomyopathy characterized by cristae abnormalities and cardiac arrhythmias in a canine model. This natural animal model of mitochondrial cardiomyopathy provides a large animal model with which to study the development and progression of disease as well as genotypic phenotypic relationships.

## 1. Introduction

The *QIl1* gene produces a protein component of the Mitochondrial Contact Site and Cristae Organizing System (MICOS), a complex made up of seven core subunits that form and stabilize mitochondrial cristae junctions and determine the placement, distribution and copy number of the cristae in the mitochondria [1,2]. The mitochondrial cristae contain the respiratory chain complexes needed for oxidative phosphorylation and the production of a significant amount of cellular ATP [3]. Loss of QIL1 has been associated with the loss of cristae junctions, cristae rearrangement into stacks of concentric membranes, and reduced cellular respiration [4]. The important role of QIL1 in cellular energy production would suggest that a dysfunctional protein would have a likely impact on organs with the highest energy needs, including the liver, brain, skeletal muscle and heart [5]. DNA variants in the *QIL1* gene have previously been identified in infants with hepatocellular dysfunction, mitochondrial encephalopathy, and in one patient, hypertrophic cardiomyopathy [1,6,7]. Cardiac arrhythmias in the absence of structural cardiac changes have not yet been reported.

Here, we report the association of a novel variant in the *QIL1* gene with familial cardiac arrhythmias in the Rhodesian Ridgeback dog. We have previously reported this canine model of familial arrhythmias and sudden death [8]. Affected dogs had cardiac arrhythmias but had no evidence of cardiac hypertrophy, myocardial dysfunction or abnormal cardiac histologic findings. Skeletal muscle was found to be consistent with the previously identified mitochondrial abnormalities in human patients with *QIl1* variants [6]. We report here the first example of a *QIL1* variant associated with a mitochondrial arrhythmic cardiomyopathy.

## 2. Materials and Methods

This study was conducted in accordance with the guidelines of the North Carolina State University Institutional Animal Care and Use Committee (IACUC, 17-168-0).

We previously reported an extended family of Rhodesian Ridgebacks with juvenile cardiac arrhythmias that occasionally resulted in sudden cardiac death [8]. Affected dogs were noted to have ventricular arrhythmias (Figure 1) most commonly between seven and twelve months of age. Atrial premature beats and second-degree atrioventricular block were noted as well, although much less commonly. 

A trio that included an apparently unaffected 70-month-old sire, an apparently unaffected 58-month-old dam and an affected 13-month-old female offspring was selected from the family for whole genome sequencing. The affected phenotype was determined by cardiac evaluation including a 24 h ambulatory electrocardiogram (Holter monitor) with at least 50 ventricular premature complexes/24 h and echocardiogram by a board-certified veterinary cardiologist that indicated no structural reason for the arrhythmia [9,10]. The sire and dam were considered to be unaffected based on ambulatory electrocardiograms with one, and zero ventricular premature beats, respectively, per 24 h at the time of evaluation. The sire and dam did not have a history of ventricular ectopy or previous evidence of cardiac disease; however, a previous mating had produced a female offspring who developed ventricular arrhythmias at 10 months of age. Three male offspring without evidence of ventricular ectopy were also produced.

Approximately three milliliters of whole blood was collected in an EDTA tube from each animal. Genomic DNA was extracted using the QIAmp DNA Blood Kit standard protocol (Qiagen, Germantown, MD, USA). Three micrograms of DNA from each dog was submitted for library preparation and whole genome sequencing, using a 150 base pair (bp) paired-end read configuration on an Illumina HiSeq 4000 high-throughput sequencing system (Genewiz LLC, South Plainfield, NJ, USA).

Variant calling from next-generation sequencing data was performed using a standardized bioinformatics pipeline for all samples, as described previously [11]. Sequence reads were trimmed using Trimmomatic 0.32 to a minimum phred-scaled base quality score of 30 at the start and end of each read, with a minimum read length of 70 bp, and aligned to the canFam3 reference sequence using BWA 0.7.13 [12,13]. Aligned reads were prepared for analysis using Picard Tools 2.8 (http://broadinstitute.github.io/picard) and GATK 3.7 following best practices for base quality score recalibration and indel realignment (Broad Institute, Cambridge, MA, USA) [14,15,16]. Variant calls were made using GATK’s HaplotypeCaller walker, and variant quality score recalibration (VQSR) was performed using sites from dbSNP 146 and the Illumina 174K CanineHD BeadChip as training resources. A VQSR tranche sensitivity cutoff of 99.9% to SNPs and 99% to indels was used for downstream analyses; genotype calls with a phred-scaled quality score < 20 were flagged but not removed from the variant callset.

Variants in the trio of dogs were filtered for polymorphisms consistent with both an autosomal dominant and recessive inheritance pattern. The resulting variants were then filtered against a previously established database of variants from 247 non-Rhodesian Ridgeback dogs of 43 different dog breeds. Any variants with a minor allele frequency greater than 1% in the non-Rhodesian Ridgebacks were removed. The remaining variants were categorized by Variant Effect Predictor 91 (https://useast.ensembl.org/info/docs/tools/vep/index.html) and prioritized by their functional impact (e.g., stop codon, frameshift, change in amino acid, etc.) [17]. They were manually curated for potential cardiac involvement of the gene (cardiac expression, encoding for cardiac channel proteins, sarcomeric proteins, cytoskeletal or mitochondrial proteins, previous association with cardiomyopathy or arrhythmic disease). Missense variants were evaluated for genomic functional significance with Polyphen (http://genetics.bwh.harvard.edu/pph2/), SIFT (http://sift.jcvi.org/) and Provean (http://provean.jcvi.org/index.php).

High impact variants (missense, stop/start gained or lost, inframe deletion, frameshift) with potential cardiac involvement were evaluated for previous identification in the canine population in the DogSD (http://bigd.big.ac.cn/dogsdv2/) SNP database. DNA SNPs that were not previously reported were pursued with Sanger Sequencing in five affected and five apparently unaffected dogs, and assessed for statistical association to the arrhythmia with a Fisher’s exact test. Variants that were significantly associated with disease (*p*-value of <0.05) were evaluated by Sanger Sequencing of eight additional family members of the trio used for whole genome association, 106 affected Rhodesian Ridgebacks and 120 unaffected dogs of 47 different dog breeds maintained in an archive at the North Carolina State University College of Veterinary Medicine. The variants were tested for allelic association with the arrhythmia using a Fisher’s exact test. A *p*-value of <0.05 was considered significant.

To determine the impact of the mutation at the skeletal muscular level, a biopsy of the biceps femoris muscle was performed under general inhalational anesthesia on a 15-month-old affected female Rhodesian Ridgeback dog homozygous for a *QIL1* variant. Following collection, the samples were immersion-fixed in Karnovsky’s fixative. Samples were evaluated with electron microscopy. For comparison, archived control muscle from a large mixed breed dog were similarly processed.

## 3. Results

Affected dogs demonstrated ventricular and/or supraventricular tachycardia and occasional atrioventricular block that developed between 7–12 months of age (Figure 1). Analysis of the whole genome sequences identified 271,877 variants consistent with an autosomal recessive pattern. Variants were the filtered to identify those that were in the affected Rhodesian Ridgeback, sire and dam, and not in >1% of the alleles in the non-Rhodesian Ridgeback dog database. This reduced the number of variants to 32,599 that would be consistent with an autosomal recessive pattern. Similarly, analysis identified 1,080,041 variants consistent with an autosomal dominant pattern, and 239,780 remained after filtering.

The majority of the variants were predicted to be of low or moderate impact (synonymous, 3′ or 5′ untranslated regions, upstream or downstream of a gene, intronic), and were not pursued for additional evaluation. One hundred and seven of the variants were predicted to be of higher impact, including five variants predicted to create a frameshift, 10 predicted to create either an inframe deletion or insertion, twenty splice site variants and seventy-two missense mutations. Thirteen of these higher impact variants were predicted to have cardiac involvement, including variants identified in the *ADCY3*, *AGRN*, *BSCL2*, *FASTKD3*, *FAT1*, *HCN4*, *LAMA4*, *MYO9B*, *PIEZO2*, *PRDM8*, *QIL1*, *SMTNL1* and *SORBS2* genes. Each of these variants was evaluated by Sanger Sequencing of ten (five affected, five unaffected) additional dogs, and a Fisher’s Exact test was performed to test for association of the variant to the arrhythmia.

Only one variant, a single missense variant ENSCAFG00000018796 g.54343438 G>A in exon four of the C19orf70 (*QIL1*) gene, had a statistical association with the arrhythmia (*p* = 0.04) (Table 1) (Figure 2).

Additionally, the variant was significantly associated with the arrhythmia in the Rhodesian Ridgeback family (*p* = 0.001) and was identified as homozygous in the affected offspring and heterozygous in both of the parents. The variant was strongly associated with the arrhythmic disease (Fisher’s exact *p* < 0.0001) in the population of 106 affected Rhodesian Ridgebacks compared to the control population of non-Ridgeback dogs. The SNP was not identified in DogSD (http://bigd.big.ac.cn/dogsdv2/) as a known canine SNP.

The *QIL1* variant was predicted to change the amino acid produced at this location from a highly conserved Glycine to Serine, and was predicted to be a deleterious change by all three variant prediction algorithms. Polyphen predicted the variant to be likely damaging (score of 1; scores of 0.85–1 predicted to be deleterious); SIFT predicted it to be deleterious (score of 0; scores of 0–0.05 predicted to be deleterious) and Provean predicted it to be a deleterious change (score of −3.5; scored of −2.5 or less predicted to be deleterious).

Ultrastructural analysis of the biceps femoris muscle from an affected homozygous Ridgeback revealed hyperplastic mitochondria, cristae rearrangement including irregular membranous swirls, and electron dense inclusions and lipid bodies (Figure 3A,B and Figure 4) consistent with pathologic changes described in the similar human disorder [6] and mouse model [4]. Large mitochondria spanned over 3 sarcomeres (Figure 3B). In contrast, mitochondria from control muscle were variable in size, and the largest spanned up to 1 sarcomere.

## 4. Discussion

In the study presented here, we report a mitochondrial arrhythmic cardiomyopathy associated with a DNA variant in the *QIL1* gene in a spontaneous canine model. We have previously reported that this cardiomyopathy was characterized by familial ventricular arrhythmias and sudden cardiac death in a population of young Rhodesian Ridgeback dogs [8]. Here we report the association of this cardiomyopathy and characteristic mitochondrial abnormalities in skeletal muscle with the *QIL1* variant.

Mitochondrial cardiomyopathies can be associated with either nuclear or mitochondrial variants [18]. QIL1 is a nuclear protein that is imported into the mitochondria and is important for proper assembly of the MICOS, which stabilizes mitochondrial cristae junctions and determines the placement and distribution of mitochondrial cristae [1,2]. QILI depletion has been associated with enlarged mitochondria, increased lipid droplets, cristae morphologic defects including a curvilinear pattern and concentric stacking of the inner mitochondrial membrane, and reduced mitochondrial respiration [1,2,4,7]. Since mitochondrial respiration is critical for the generation of ATP via electron transport and oxidative phosphorylation systems, organ systems that have particularly high energy demands, including the brain, liver, skeletal muscle and the heart, are most likely to be impacted by mitochondrial dysfunction [5]. *QILI* variants have been previously associated with the development of infantile encephalopathy, liver dysfunction and in one patient, hypertrophic cardiomyopathy [1,6,7]. Since the heart is one of the most energy demanding organs, mitochondrial diseases often preferentially impact the heart [18], and it has been estimated that cardiac involvement including structural and/or arrhythmic abnormalities can occur in 20–40% of children with mitochondrial disease [5,19,20]. We report here on young Rhodesian Ridgeback dogs with familial arrhythmias including supraventricular and ventricular tachycardia and atrioventricular block. We have previously reported the absence of structural myocardial involvement in this model [8]. These arrhythmic findings are consistent with those previously reported in mitochondrial cardiomyopathies [18]. It has been hypothesized that the development of these arrhythmias in mitochondrial cardiomyopathies may be associated with dysfunctional mitochondrial respiration and decreased ATP synthesis and its impact on cardiovascular action potential development, myocardial conduction [21] and electrical stability.

## 5. Conclusions

In conclusion, we identify here a variant in the *Q1l1* gene resulting in a mitochondrial cardiomyopathy characterized by cristae abnormalities and cardiac arrhythmias in a canine model. This natural animal model of mitochondrial cardiomyopathy provides a large animal model with which to study the development and progression of this disease as well as our understanding of genotypic phenotypic relationships. Additionally, it serves as model with which to study the impact of medical management on mitochondrial dysfunction.

## Figures and Tables

**Figure 1 genes-10-00168-f001:**
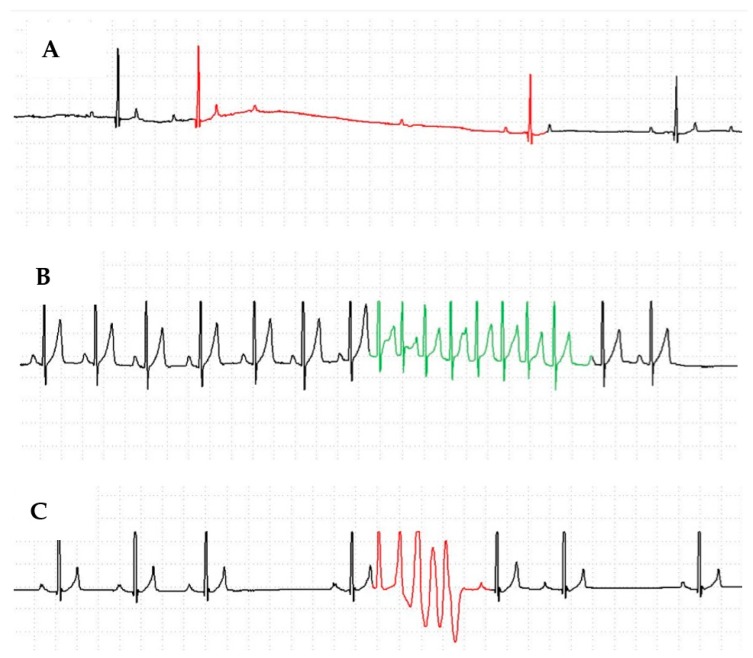
Affected Rhodesian Ridgebacks were observed to have three different cardiac arrhythmias. (**A**) Second degree atrioventricular block. (**B**) Sinus rhythm with supraventricular tachycardia. (**C**) Sinus rhythm with ventricular premature complexes.

**Figure 2 genes-10-00168-f002:**
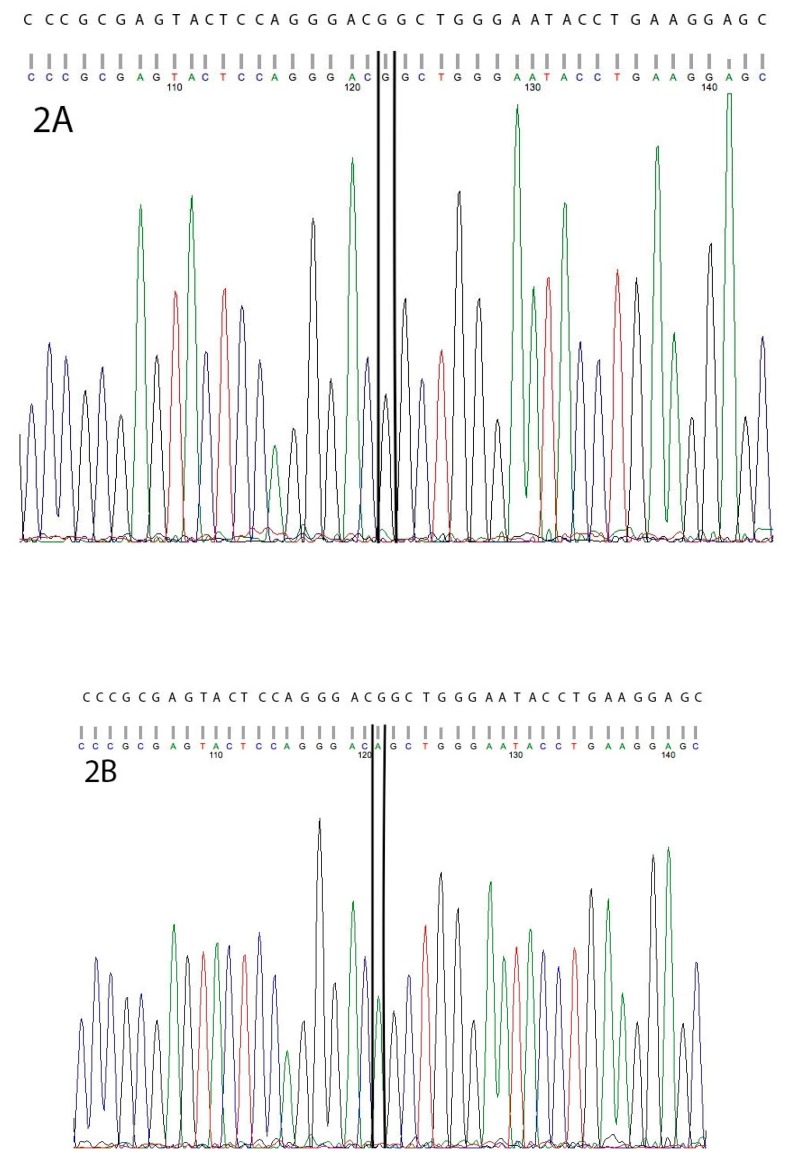
(**A**) Normal genetic sequence. (**B**) Sequence of a Rhodesian Ridgeback with a homozygous variant (G/A) outlined by black box. Reference sequence (ENSCAFG00000018796) alignment is above the DNA chromatogram.

**Figure 3 genes-10-00168-f003:**
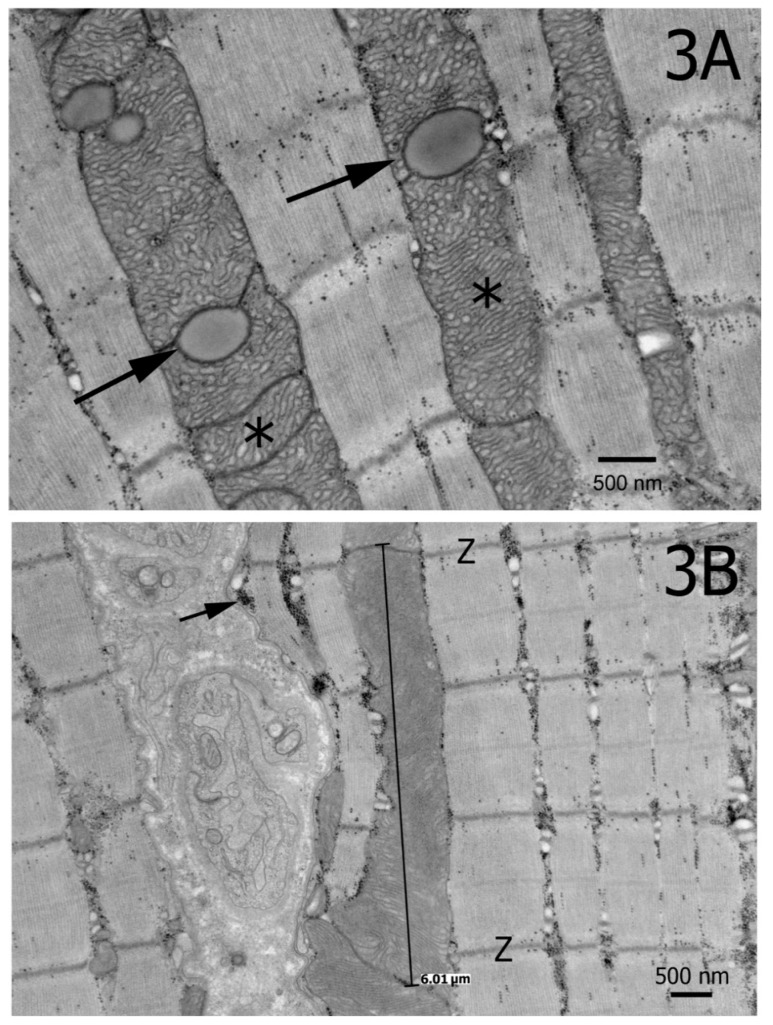
(**A**) Electron micrograph showing hyperplastic mitochondria with irregular cristae (*). Arrows highlight lipid bodies. (**B**) A large mitochondrion spans over 3 sarcomeres. Membrane proliferation in swirls and electron dense inclusions (arrow) were also noted. Z = z lines.

**Figure 4 genes-10-00168-f004:**
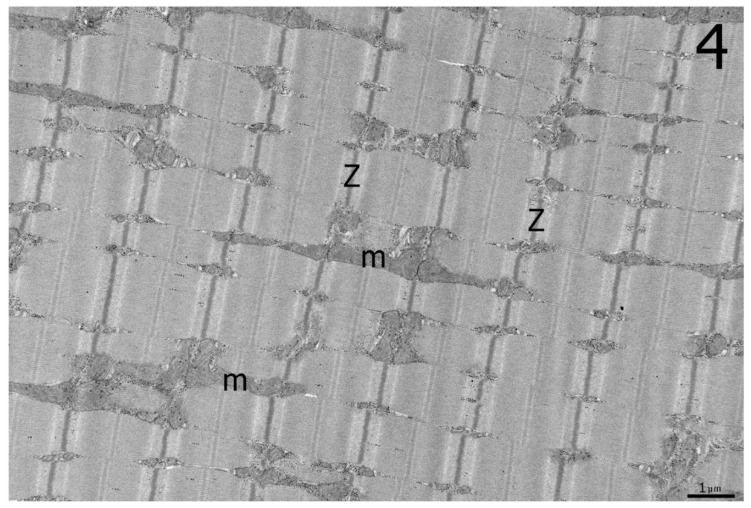
For comparison to Figure 3, an electron micrograph from archived control dog muscle shows variability in the size of mitochondria with the largest mitochondrion spanning 1 sarcomere Z = z line, m = mitochondria.

**Table 1 genes-10-00168-t001:** Variants evaluated by Sanger sequencing. Gene, variant location, effect and statistical association by Fisher’s exact test are provided.

Gene	Variant	Effect	*p* Value
ADCY3	ENSCAFG00000004090	Missense/Splice site	NP
	g.19164291 G>A		
AGRN	ENSCAFG00000019342	Missense	0.25
	g.56260122 C>A		
BSCL2	ENSCAFG00000023629	Initiator codon variant	0.62
	g.53960802 A>G		
FASTKD3	ENSCAFG00000010129	Missense	0.63
	g.6105469 C>T		
FAT1	ENSCAFG00000007273	Missense	0.16
	g.44199325 G>T		
HCN4	ENSCAFG00000031809	Frameshift	NP
	g.36680881_36680885del		
LAMA4	ENSCAFG00000004043	In frame deletion	NP
	g.68553193_68553195del		
MYO9B	ENSCAFG00000015532	Missense	0.35
	g.45524791 C>T		
PIEZO2	ENSCAFG00000018761	Frameshift	NP
	g.76508892_76508892insG		
PRDM8	ENSCAFG00000008881	Inframe insertion	NP
	g.4450426_4450427insG		
C19orf70	ENSCAFG00000018796	Missense	0.04
	g.54343438 G>A		
SMTNL1	ENSCAFG00000007843	Missense	>0.99
	g.38627683 C>T		
SORBS2	ENSCAFG00000007475	Missense	0.035
	g.45035993 G>A		

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
