# Peer review of "A QIL1 Variant Associated with Ventricular Arrhythmias and Sudden Cardiac Death in the Juvenile Rhodesian Ridgeback Dog"

_genes, 2019, doi:10.3390/genes10020168_

Round 1

Reviewer 1 Report

An interesting and well written presentation of the discovery of a variant in QIL1 and its link to cardiac arrhythmia and skeletal muscle mitochondrial abnormalities in the Rhodesian Ridgeback dog. A novel finding and I have no issues with the science as presented, though I am not a bio-informatician.

My main concern is the lack of clarity of the cardiac phenotype in the paper. Was there hypertrophy/ cardiomyopathy ? Echocardiography/histology? At what age do these dogs die, how many die and from what? (some die a sudden death, do others die young from cardiac failure?)

It may be that data which is presented in other work, or available from the other dogs, can clear this up, and if so I urge the authors to do so.

Author Response

An interesting and well written presentation of the discovery of a variant in QIL1 and its link to cardiac arrhythmia and skeletal muscle mitochondrial abnormalities in the Rhodesian Ridgeback dog. A novel finding and I have no issues with the science as presented, though I am not a bio-informatician.

My main concern is the lack of clarity of the cardiac phenotype in the paper. Was there hypertrophy/ cardiomyopathy ? Echocardiography/histology? At what age do these dogs die, how many die and from what? (some die a sudden death, do others die young from cardiac failure?)

It may be that data which is presented in other work, or available from the other dogs, can clear this up, and if so I urge the authors to do so.

Thank you for the reviewer’s time and comments on our manuscript. We apologize for the brevity of the phenotypic description. We have added more about the phenotype to the manuscript. We had published this previously* and were being overly careful about including it here.

Briefly, these affected Ridgeback dogs had no evidence of hypertrophy or myocardial dysfunction. Echocardiography and histology were normal. The electron microscopy findings were the only abnormality noted on pathologic evaluation. The dogs generally die suddenly between 7 and 14 months of age.

*Meurs, K.M.; Weidman, J.A.; Rosenthal, S.L.; Lahmers, K.K.; Friedenberg, S.G. Ventricular arrhythmias in Rhodesian Ridgebacks with a family history of sudden death and results of a pedigree analysis for potential inheritance patterns. J. Am. Vet. Med. Assoc. 2016, 248, 1135-1138.

Reviewer 2 Report

In the manuscript titled “A QIL1 Variant Associated with Ventricular Arrhythmias and Sudden Cardiac death in the juvenile Rhodesian Ridgeback Dog”, Meurs and his co-workers used the whole genome sequencing technology to screen for mutations a Rhodesian Ridgeback Dog spontaneous model with ventricular arrhythmias and sudden cardiac death. They identified several high impact variants in genes predicted to have cardiac involvement among which is a new missense variant in QIL1, a gene associated with MICOS disassembly and the onset of several mitochondrial diseases. Based on an association statistical study and an electron microscopy analysis of the skeletal muscular tissue of the affected animals, the authors came to the conclusion of a cause-effect relationship between the QIL1 variant and the arrhythmic phenotype in the Rhodesian Ridgeback Dog. Although the author´s findings sound scientifically interesting, there are some major issues concerning the experiment design and the statistical analysis that might question the quality of the work:

Major comments:

Line 108: “…assessed for statistical association to the arrythmias…”, did the authors base their statistical analysis on the 10 dogs only (five affected+5 unaffected)? If so, this number might question the significance of the statistics. 10 animals seem to be insufficient to base on a statistical association analysis.

Figure 3: Please add the total number of mitochondria analysed and the number of abnormal mitochondria out of this total for figure3A and 3B.

Line 145: How many family members were screened? How did the authors establish an association study based on samples belonging to the same family? Related samples are avoided in the association studies.

Line 147: Why did the authors use the non-Ridgebacks dogs as control population? Why not the unaffected Ridgebacks dogs? It would be even genetically rational working with a control population from the same race.

Table 1: Please apply the mutation nomenclature guidelines when describing each of the identified variants. Eg.: (c.30-1G>A) in C19orf70 (QIL1). Please use the full nomenclature (with the amino acid change and position) when mentioning the newly identified QIL1 variant.

Line 194: “resulting ….” How did the authors establish this cause-effect relationship between the QIL1 variant and the phenotype if no functional study was performed to this variant? Same as in line 115, the phenotype could be due to a cumulative effect of several variants together. In order to correlate the phenotype with the QIL1 variant, it would be interesting if the authors perform a functional study expressing the QIL1 variant in cardiomyocytes and skeletal muscle derived cells in order to assess its functional effect.

Minor comments:

Figure 1: Since the figure is part of the authors work and is not presented previously in reference 8, it would be more appropriate if they move it to the results section although the elctrophysiological characterisation of the animal model was performed in previous works. Else, authors could just mention the ECG phenotype without including the figure1.

Line 122: “After …….32,599 variants remained” please reformulate the sentence to make it easier to understand.

Figure 2: The resolution of the chromatograms should be improved. Please reduce the height of chromatogram peaks and increase the space between the nucleotides to make it easier to read.

Figure 3: The figure legend: “….dark electron incusions” did the authors mean “…..dark electron inclusions”?

Author Response

In the manuscript titled “A QIL1 Variant Associated with Ventricular Arrhythmias and Sudden Cardiac death in the juvenile Rhodesian Ridgeback Dog”, Meurs and his co-workers used the whole genome sequencing technology to screen for mutations a Rhodesian Ridgeback Dog spontaneous model with ventricular arrhythmias and sudden cardiac death. They identified several high impact variants in genes predicted to have cardiac involvement among which is a new missense variant in QIL1, a gene associated with MICOS disassembly and the onset of several mitochondrial diseases. Based on an association statistical study and an electron microscopy analysis of the skeletal muscular tissue of the affected animals, the authors came to the conclusion of a cause-effect relationship between the QIL1 variant and the arrhythmic phenotype in the Rhodesian Ridgeback Dog. Although the author´s findings sound scientifically interesting, there are some major issues concerning the experiment design and the statistical analysis that might question the quality of the work:

Major comments:

Line 108: “…assessed for statistical association to the arrythmias…”, did the authors base their statistical analysis on the 10 dogs only (five affected+5 unaffected)? If so, this number might question the significance of the statistics. 10 animals seem to be insufficient to base on a statistical association analysis.

We agree that this is a small number. We used 10 only as the first indication that a variant could be associated with disease. If the variant was not determined to appear to be associated with disease in the 5 affected and 5 unaffected dogs we would not pursue it further. If it did appear to be associated with disease we would evaluate it for association within the family and the larger population of affected and unaffected dogs. The QIL1 variant was the only variant that appeared to be associated with disease in the 10 dogs and continued to be associated with disease as we looked at larger numbers of dogs (lines 138-141 and lines 144-149).

Figure 3: Please add the total number of mitochondria analysed and the number of abnormal mitochondria out of this total for figure3A and 3B.

We did not perform quantitative studies on the mitochondria. Subjectively in the EM figure, abnormal mitochondria were numerous

Line 145: How many family members were screened?

The family included eleven family members including 4 offspring from a second mating and three offspring of mating the sire to different females. We have clarified this in the paper.

 How did the authors establish an association study based on samples belonging to the same family? Related samples are avoided in the association studies.

For the sake of association within the family we did a Fischer’s exact test to assess association of the variant to the affected individuals in a similar fashion to how the variant was assessed in the overall population. Although we agree that this is not typical of an Association study, it was simply to continue demonstrating the relationship of the variant to affected dogs.

Line 147: Why did the authors use the non-Ridgebacks dogs as control population? Why not the unaffected Ridgebacks dogs? It would be even genetically rational working with a control population from the same race.

We agree that it would have been ideal to use unaffected Ridgebacks for the study. However, this is a fairly recently described familial disease. It is a disease of variable expression and there is some suggestion that expression of the disease may decrease with maturity of the dog as energy demands decrease. We were concerned that we could accidentally include Ridgeback dogs with very mild disease expression into an “unaffected” population and could miss detection of the variant.  Therefore, we filtered against non-Ridgeback dogs.

Table 1: Please apply the mutation nomenclature guidelines when describing each of the identified variants. Eg.: (c.30-1G>A) in C19orf70 (QIL1). Please use the full nomenclature (with the amino acid change and position) when mentioning the newly identified QIL1 variant.

We have added the correct nomenclature to the text and altered Table 1 to reflect the correct nomenclature.

Line 194: “resulting ….” How did the authors establish this cause-effect relationship between the QIL1 variant and the phenotype if no functional study was performed to this variant? Same as in line 115, the phenotype could be due to a cumulative effect of several variants together. In order to correlate the phenotype with the QIL1 variant, it would be interesting if the authors perform a functional study expressing the QIL1 variant in cardiomyocytes and skeletal muscle derived cells in order to assess its functional effect.

In this case we correlated the QIL1 variant to the phenotype based on a series of over lapping associations: 1) statistical association of variant to the arrhythmic phenotype in affected dogs, 2) identification of mitochondrial changes in dogs with the QIL1 variant that are consistent with those previously described in humans and animal models with QIL1 variants, 3) identification of cardiac arrhythmias that have previously associated with cases of mitochondrial dysfunction.

We agree that future studies expressing the variant in cell lines and assessing function will be beneficial to further understand this variant.

Minor comments:

Figure 1: Since the figure is part of the authors work and is not presented previously in reference 8, it would be more appropriate if they move it to the results section although the elctrophysiological characterisation of the animal model was performed in previous works. Else, authors could just mention the ECG phenotype without including the figure1.

We moved Figure 1 to the Results section. Thank you

Line 122: “After …….32,599 variants remained” please reformulate the sentence to make it easier to understand.

We have altered the sentence, we agree that it was not well written.

Figure 2: The resolution of the chromatograms should be improved. Please reduce the height of chromatogram peaks and increase the space between the nucleotides to make it easier to read.

We have improved the resolution of the figure. We apologize for the quality of the previous version.

Figure 3: The figure legend: “….dark electron incusions” did the authors mean “…..dark electron inclusions”?

Yes, thank you for catching this. We have corrected this.

Round 2

Reviewer 2 Report

The authors answered to most of the comments raised during the first review round. Nonetheless, there are still some points that should be clarified:

Major comments:

-The authors answered the first major comment in the previous report referring to (lines 138-141 and lines 144-149), however these lines are not related to the raised point.

-Figure 2: Please provide the sequence alignment in support of the variant locus conservation.

-Figure 3: If no mitochondria quantification was performed, a control electron micrograph would support your findings. Please provide this data.

-Table 1: in the revised version, the p value corresponding to the QIL1 variant changed to 0,004 instead of 0,04 in the old version. Please give a justification. 

Please keep the variant effect colon along with the new added data in the table. It will give a clearer idea: Gene/ Variant/ Effect/ p value. 

Minor comments:

-Line 122: "Sample were evaluated with and electron microscopy", did the author mean "Sample were evaluated with and electron microscopy"

-Line 134:  "were than...." , did the authors mean "were then..."

-Line 167: please review the sentence again. there are some missing words.

Author Response

We appreciate the opportunity to revise this manuscript. We apologize for missing the typos and errors in the previous version.

The authors answered to most of the comments raised during the first review round. Nonetheless, there are still some points that should be clarified:

Major comments:

-The authors answered the first major comment in the previous report referring to (lines 138-141 and lines 144-149), however these lines are not related to the raised point.

We apologize for the confusion. The original question was Line 108: “…assessed for statistical association to the arrhythmias…”, did the authors base their statistical analysis on the 10 dogs only (five affected+5 unaffected)? If so, this number might question the significance of the statistics. 10 animals seem to be insufficient to base on a statistical association analysis.

We did start with 10 dogs (5 affected and 5 unaffected). If there was no clear association (variant was often not in all the affected or appeared in several of the unaffected) we did not pursue it farther.

-Figure 2: Please provide the sequence alignment in support of the variant locus conservation.

We have added the sequence above each figure.

-Figure 3: If no mitochondria quantification was performed, a control electron micrograph would support your findings. Please provide this data.

We have added an additional control figure from a normal dog (Figure 4)

-Table 1: in the revised version, the p value corresponding to the QIL1 variant changed to 0,004 instead of 0,04 in the old version. Please give a justification. 

Sincere apologies- the text remained correct at .04. Somehow we missed that there was an error in the Table stating it as 0.004. This is very sloppy mistake. We apologize for missing it and have corrected it.

Please keep the variant effect colon along with the new added data in the table. It will give a clearer idea: Gene/ Variant/ Effect/ p value. 

We have added this to the table

Minor comments:

-Line 122: "Sample were evaluated with and electron microscopy", did the author mean "Sample were evaluated with and electron microscopy"

Yes, this is correct. We apologize and we have corrected this.

-Line 134:  "were than...." , did the authors mean "were then..."

Yes, this is correct, we have corrected this

-Line 167: please review the sentence again. there are some missing words.

We have corrected this sentence, thank you